# Identification the Cross-Reactive or Species-Specific Allergens of *Tyrophagus putrescentiae* and Development Molecular Diagnostic Kits for Allergic Diseases

**DOI:** 10.3390/diagnostics10090665

**Published:** 2020-09-02

**Authors:** Ching-Hsiang Yu, Jaw-Ji Tsai, Yi-Hsueh Lin, Sheng-Jie Yu, En-Chih Liao

**Affiliations:** 1Department of Medicine, MacKay Medical College, New Taipei City 252, Taiwan; 110310028@live.mmc.edu.tw; 2Division of Allergy, Immunology & Rheumatology, Department of Internal Medicine, Asia University Hospital, Taichung 413, Taiwan; d00010@auh.org.tw; 3Institute of Clinical Medicine, National Yang Ming University, Taipei 112, Taiwan; yi.xeu.lin@gmail.com; 4Department of Medical Education and Research, Kaohsiung Veterans General Hospital, Kaohsiung 813, Taiwan; jim0929@msn.com; 5Department of Medicine, Mackay Medical College No. 46, Sec. 3, Zhongzheng Rd., Sanzhi Dist., New Taipei City 252, Taiwan

**Keywords:** *Dermatophagoides pteronyssinus*, *Tyrophagus putrescentiae*, storage mite, mite allergens, allergic diseases, diagnosis technology

## Abstract

Mite allergens are considerable factors in the genesis of allergic diseases. The storage mite *Tyrophagus putrescentiae* (Tp) appears in contaminated foods and household surroundings. The current diagnostic tools for Tp allergy are mostly based on crude extracts and still contain shortcomings. This study aimed to investigate the immunoglobulin E (IgE)- responsiveness profiles of Tp-allergic patients and develop a molecular diagnostic method using recombinant allergens. Allergenic components were characterized as cross-reacting or species-specific allergens, in which the effective combinations of recombinant allergens were developed and analyzed in terms of the prediction accuracy for clinical diagnosis. Seven recombinant allergens were cloned and generated to detect the IgE responsiveness of the Tp allergy. A survey on the prevalence of mite allergy showed there were higher sensitizations with IgE responsiveness to house dust mites (HDM) (78.9–80.9%) than to storage mites Tp (35.6%). Prevalence of sensitization to Tp was higher in elderly subjects. The principal IgE-binding components of Tp were Tyr p 1, Tyr p 2 and Tyr p 3. Prediction accuracy for Tp allergy by IgE-responsiveness combination D (Tyr p 1, Tyr p 2 & Tyr p 3) was with high precision (100%). Avoiding the cross-reactivity of *Dermatophagoides pteronyssinus*, the prediction accuracy of IgE-responsiveness combination H+ (Tyr p 1, Tyr p 2, Tyr p 3, Tyr p 7, Tyr p 8, Tyr p 10 & Tyr p 20) was suitable for Tp-specific diagnosis. Panels of Tp allergens were generated and developed a diagnostic kit able beneficial to identify IgE-mediated Tp hypersensitivity.

## 1. Introduction

There is increasing evidence that the frequency of allergic diseases such as asthma, rhinitis and dermatitis has increased worldwide over the last few decades and becomes a global public health problem [1,2,3]. The indoor inhalant allergens of mites have become the most important triggers of IgE-mediated responses and allergic diseases in genetically predisposed patients [4,5]. The most prevalent indoor allergen sources are domestic mites [6], they majorly can be divided into two categories: pyroglyphid and non-pyroglyphid mites—commonly referred to as house dust mites and storage mites, respectively [6,7].

House dust mites—especially *Dermatophagoides pteronyssinus*—are considered to be an important source for allergen sensitization and a major risk factor for allergic symptoms [8,9]. *D. pteronyssinus* occurs at high levels of infestation in houses around the world and with high IgE frequency, which the mite extracts have indicated over 36 different allergenic components can induce IgE in patients and about 32 allergens been identified [9]. In view of allergenic potential and clinically relevant, the storage mite *Tyrophagus putrescentiae* has been reported to cause anaphylaxis through accident ingestion of mite-contaminated foods and induce allergic respiratory symptoms after occupational or inhabited exposure [10,11]. At least 20 kinds of IgE-binding allergenic components of *T. putrescentiae* have been detected in allergic patient’s sera [9] but not so many allergenic components like *D. pteronyssinus* have been cloned, characterized and clarified their IgE-binding frequency or biochemical function. According to the information of World Health Organization/International Union of Immunological Societies (WHO/IUIS) Allergen Nomenclature Sub-Committee (http://www.allergen.org/), a total of nine allergens of *T. putrescentiae* have been registered including Tyr p 2, Tyr p 3, Tyr p 8, Tyr p10, Tyr p 28, Tyr p 34, Tyr p 35 and Tyr p 36 [12]. The characterization of *T. putrescentiae* allergens is essential for the development of a diagnostic method and therapeutic agents against mite-induced allergic disorders.

Cosensitization can be observed in some patients who are cosensitized independently to two or more different allergen sources in nature [13]. The cosensitization to storage mite *T. putrescentiae* and house dust mite *D. pteronyssinus* is a frequent finding of allergic subjects in urban or humid areas [14,15]. Cross-reactivity occurs because of shared similar IgE-binding epitopes among different allergens [16]; thus, these shared epitopes in closed or distantly related species of allergens lead to the development of allergic symptoms [16,17]. Our previous study showed considerable cross-reactivity in Group 2 allergen between *T. putrescentiae* and *D. pteronyssinus*, which is one of the most prevalent allergens among both [18]. However, the levels of cross-reactivity among other groups of allergen between *T. putrescentiae* and *D. pteronyssinus* still have not been well investigated. Evaluating the cross-reactivity between both will provide more information in which allergenic components belong to cross-reacting allergens or species-specific allergens, which is useful for allergen categorization of diagnostic agents or immunotherapeutic agents. The availability of species-specific and cross-reactive marker components creates the platform for more informative diagnostics.

Currently, the most commonly used standard approaches in clinical diagnosis for mite sensitization are crude extract-based methods, such as skin prick test, provocation test and serology assay [19,20]. These diagnostic tools based on crude extracts still contain some disadvantages and insufficiencies such as difficult to be consistent between different production batches, remain undefined mixtures of allergenic and nonallergenic materials [21,22]. There is a trend toward utilizing the recombinant allergens for the molecular-based diagnostic methods [23]. Most recombinant allergens can be expressed in large amounts in *Escherichia coli* or insect cells at low cost without risk of containing infectious materials [24]. The availability of ImmunoCAP immuno solid-phase allergen chip (ISAC^®^) microarray-based determination of IgE against *D. pteronyssinus* recombinant allergens provides molecular IgE-reactivity profiles is associated with clinical manifestations in mite-allergic patients [25].

To date, no relevant studies focusing on the development of differential diagnostic kit for *T. putrescentiae* allergy have been investigated. The aim of this study was to investigate the IgE-responsiveness profiles in *T. putrescentiae*-allergic patients. Evaluating the cross-reactivity of these allergenic components between *T. putrescentiae* and *D. pteronyssinus*, to characterize allergens belong to cross-reactive allergens or species-specific allergens, which we can develop the effective agent combination for clinical diagnosis.

## 2. Materials and Methods

### 2.1. Study Subjects

A total of 298 allergic subjects (167 males and 131 females) who attended the Clinic in the Division of Allergy, Immunology and Rheumatology at Taichung Veterans General Hospital (TCVGH) were recruited for this study. These allergic patients were diagnosed with allergies by a specialist attending physician if they had at least one of the following history: asthma, rhinitis, atopic dermatitis or eczema. The collection of blood samples from patients was approved by the Institutional Review Board, TCVGH (TCVGH IRB No. C07126). Written informed consent was obtained from each participant before being enrolled in the study. Blood samples were drawn, and sera were stored for the measurement of specific IgE to *D. pteronyssinus*, *D. farinae* and *T. putrescentiae*.

### 2.2. Preparation of Mite Crude Extracts

Two kinds of mite crude extracts were used in this study. The crude extracts of *D. pteronyssinus* were prepared from lyophilized whole mite bodies purchased from Allergon (Angelholm, Sweden). The *T. putrescentiae* mites were cultured in our laboratory with a medium consisting of yeast extract and mouse chow, then mite bodies were separated from the medium. The mite collection is described in our previous study [26]. In brief, separation of the mites from the medium was achieved by gently stirring the medium with a glass rod following overnight culture, and then the mites migrated to the cover would be collected. This collection included a large proportion of mite bodies. Frozen *T. putrescentiae* mites were homogenized and extracted with phosphate buffer saline (PBS, pH 7.2). Protein concentration was determined by Bradford assay (Bio-Rad, Hercules, CA, USA) using bovine albumin as a standard.

### 2.3. Detection of IgE Responsiveness in Sera of Allergic Patients against Mite Crude Extracts

The IgE responsiveness against mite crude extracts in sera of allergic patients was determined by an enzyme-linked immunosorbent assay (ELISA) as previously described [27]. Briefly, the mite crude extracts used were coated separately onto wells of polyvinyl microtiter plates (Costar, Cambridge, MA, USA) by the addition of 100 μL of a 5-μg/mL solution of crude extract in coating buffer 0.1-mol/L NaHCO_3_, pH 8.4 for 4 h at RT. After blocking with 1% skimmed milk, it was then incubated for 1 h at room temperature [28]. Wells were washed with PBS containing 0.05% Tween-20 (Southern Biotech Association, Birmingham, AL, USA) (PBST). One hundred microliters of diluted human sera (1:5 dilution) were added for overnight incubation at 4 °C. Plates were washed and incubated with horseradish peroxidase (HRP)-conjugated goat anti-human IgE (1:1000) for 1 h at RT. After washing with PBST three times, the bound enzyme substrates were detected with 3, 3’, 5, 5’-tetramethylbenzidine (TMB) substrate (Invitrogen, Carlsbad, CA, USA). The reactions were stopped with 50 μL 1-N H_2_SO_4_ after 15 min, and the optical density was measured at 450 nm in a multiscan spectrophotometer (Sunrise, TECAN, Mennendorf, Switzerland). The cutoff value was mean ± 2x standard deviation as 0.23 ± 0.02 unit of the sera of 10 healthy volunteers who have no history of allergic symptoms, and samples exceeding 0.25 unit were considered as positive.

### 2.4. Preparation, Expression and Purification Recombinant Allergens of T. putrescentiae

The recombinant allergens (rTyr p 1, rTyr p 2, rTyr p 3, Tyr p 7, rTyr p 8, rTyr p 10 and rTyr p 20) of *T. putrescentiae* were prepared and used to analyze the frequency of IgE reactivity by sera from the allergic subjects. Total RNA of mites was isolated with Trizol reagent, and cDNA was synthesized with primer dT, MMLV reverse transcriptase (Gibco-BRL, New York, NY, USA). The recombinant proteins of *T. putrescentiae* allergenic components were amplified by reverse transcriptase-polymerase chain reaction (RT-PCR) from cDNA and subcloned into the expression vector. The specific primers were designed based on the accession number in the National Center for Biotechnology Information or in our previous study finding. The program of PCR was carried out at 94 °C for 10 s, 55 °C for 20 s and 72 °C for 3 min for 35 cycles. The PCR products were purified, ligated into a pQE30 vector and transformed in *Escherichia coil*-M15 strain competent cells. Transformants were selected by kanamycin (25 µg/mL) and ampicillin (100 µg/mL). Expressions of recombinant allergens were performed according to the methods described in the QIA Expressionist^TM^ Kit directions (Qiagen, Hilden, Germany). The recombinant allergens were expressed as a 6x His-tagged protein using 1-mM isopropyl-β-D-thiogalactopyranoside (Promega, Madison, WI, USA) induction. The proteins were purified by nickel-nitrilotriacetic acid agarose metal affinity column chromatography under native conditions.

### 2.5. Identification of T. putrescentiae-Specific Allergen by IgE Absorption with D. pteronyssinus

Because of the cross-reactivity between different mite species, the patients’ sera were pre-absorbed with *D. pteronyssinus* then to evaluate the IgE responsiveness for identification the species-specific components of *T. putrescentiae.* For the IgE absorption, serum samples were selected from 106 allergic subjects who had IgE antibodies against *T. putrescentiae* crude extracts using ELISA. Each serum was preincubated with 50 µL of *D. pteronyssinus* crude extracts (50 µg/mL) and left overnight at 4 °C. After the preabsorption, the subsequent procedures for determining IgE responsiveness to *T. putrescentiae* by ELISA were the same as those described previously.

### 2.6. The Frequency of IgE Responsiveness to T. putrescentiae Recombinant Allergens

Recombinant allergens of *T. putrescentiae* were prepared and used to investigate the frequency of IgE responsiveness. The IgE antibodies in serum samples were detected by ELISA. The recombinant allergens (rTyr p 1, rTyr p 2, rTyr p 3, rTyr p 7, rTyr p 8, rTyr p 10 and rTyr p 20) were coated separately onto wells of polyvinyl microtiter plates (Costar, Cambridge, Mass., USA) by the addition of 100 μL of a 5-μg/mL solution in 0.1 MNaHCO_3_ (pH 8.4) for 4 h at RT. After blocking with 1% skimmed milk, it was then incubated for 1 h at RT. Wells were washed with PBS containing 0.05% Tween-20 (Southern Biotech Association, Birmingham, Ala., USA)(PBST). The ratio 1:5 dilution of serum from each allergic patient was added in 100 µL PBS. Wells were washed again with PBST, peroxidase-conjugated anti-human IgE (1:1000 dilute in PBST) was added to the wells and incubated for 2 h at RT. After washing with PBST, the bound enzyme substrates were detected with TMB substrate (Invitrogen, USA). The reaction was stopped with 50 μL 1-N H_2_SO_4_ after 15 min, and the optical density was measured at 450 nm in a multiscan spectrophotometer (Sunrise, TECAN, Mennendorf, Switzerland). The cutoff value was mean ± 2x standard deviation as 0.23 ± 0.02 unit of the sera of 10 healthy volunteers who have no history of allergic symptoms, and samples exceeding 0.25 unit were considered as positive. The IgE-binding frequency of *T. putrescentiae* allergens among allergy patients was analyzed and compared after *D. pteronyssinus* absorption. Each serum was pre-incubated with 50 µL of *D. pteronyssinus* crude extracts (50 µg/mL) and left overnight at 4 °C. After the pre-absorption, the subsequent procedures for determining IgE responsiveness to *T. putrescentiae* allergens by ELISA were the same as those described previously.

### 2.7. Establish the Detection Combinations of IgE Responsiveness for the Diagnosis of T. putrescentiae Allergy

A total of seven allergenic components of *T. putrescentiae* were cloned and the recombinant allergens were generated for detection combinations of IgE responsiveness in sera from allergic patients. The higher frequency with IgE responsiveness of allergenic components are picked out, the higher sensitivity of detection combination for *T. putrescentiae* can be obtained. Since the cross-reactivity of *T. putrescentiae* and *D. pteronyssinus* were common in clinical practice. Sera from allergic patients with the specificity of *T. putrescentiae* allergy were confirmed after the absorption of *D. pteronyssinus*. Only those detection combinations with both high sensitivity and specificity will be used for clinical application in the future.

### 2.8. Statistical Analysis

Differences between the middle-age and elderly age population of the *T. putrescentiae* allergy were analyzed by the nonparametric alternative to the unpaired two-sample *t*-test. The *p*-values of differences less than 0.05 were considered statistically significant.

## 3. Results

### 3.1. The Prevalence of Sensitization to D. pteronyssinus, D. farinae and T. putrescentiae

The sensitization to *D. pteronyssinus*, *D. farinae* and *T. putrescentiae* of allergic patients was determined by ELISA with mite crude extracts. The results showed that the IgE reaction with positive responses of allergic subjects were 80.9% (241/298) to *D. pteronyssinus*, 78.9% (235/298) to *D. farinae* and 35.6% (106/298) to *T. putrescentiae* (shown in Figure 1A). There were more subjects sensitized to *D. pteronyssinus or D. farinae than to T. putrescentiae.* It is a possible phenomenon that multiple exposures and cosensitization to these mite allergens in nature causing free IgE filled in the sera of these allergic patients. The overlapping illustration of sensitization prevalence to these mites of these subjects, the percentage of positive IgE reactions to the dust mite (*D. pteronyssinus* and *D. farinae)* and storage mite *T. putrescentiae* were 33.6% (100/298) (Figure 1B). The prevalence of sensitization to *D. pteronyssinus* and *D. farinae*, but none to *T. putrescentiae* were 45.0% (134/298) (Figure 1B). There were 16.8% (50/298) of these subjects without IgE response against these species of mites. It seemed that most of the allergic patients from the Taichung area had higher sensitization with positive IgE responses to house dust mites (78.9–80.9%) than to storage mite *T. putrescentiae* (35.6%).

### 3.2. Differences of Mite Sensitization between Elderly and Young Adult Subjects

Differences in sensitization to *D. pteronyssinus* and *T. putrescentiae* were compared between elderly and young adult subjects. The results showed there were 84.1% (116/138) allergic subject sensitive to *D. pteronyssinus* and 81.2% (112/138) sensitive to *D. farinae* in the young adult group; there was a slightly higher prevalence of *D. pteronyssinus* sensitivity in the young adult group than in the elderly group (Figure 2). Additionally, the analysis data showed there were 44.4% (71/160) elderly subjects sensitive to *T. putrescentiae* while there were only 25.4% (35/138) young adult subjects allergic to *T. putrescentiae.* The prevalence of sensitization to *D. pteronyssinus* or *D. farinae* was slightly higher in the young adult subjects, while the prevalence of sensitization to *T. putrescentiae* was significantly higher in elderly subjects than in young adult subjects (44.4% vs. 25.4%) (*p* < 0.05).

### 3.3. Molecular Cloning and Purification of Recombinant Allergens from T. putrescentiae

The RNA of *T. putrescentiae* was isolated from alive mite bodies and the complementary DNA (cDNA) synthesized via reverse transcription [28]. The cDNA encoding for *T. putrescentiae* allergens was amplified by RT–PCR and subcloned into expression vectors. A total of seven allergenic components were expressed and purified by His-tag affinity column, including rTyr p 1, rTyr p 2, rTyr p 3, rTyr p 7, rTyr p 8, rTyr p 10 and rTyr p 20. The characteristics including biologic function, molecular weight and accession No. of allergens from *T. putrescentiae* we used in this study is listed in Table 1. The protein profiles of *T. putrescentiae* recombinant allergens on SDS-PAGE showed that the rTyr p 2 approximately at 16 kDa, the rTyr p 1, rTyr p 3, rTyr p 7 and rTyr p 8 approximately at 24–26 kDa and rTyr p 10 & rTyr p 20 approximately at 37–40 kDa (shown in Figure 3). The recombinant allergens from *T. putrescentiae* were prepared for the subsequent identification of IgE-binding frequency by the ELISA.

### 3.4. The Frequency of IgE Binding to Each Allergenic Components of T. putrescentiae before and after D. pteronyssinus Absorption

The frequency of IgE binding to the allergen can provide important information for the diagnosis of clinical allergy. A total of 106 sera with ELISA-positive reactions to *T. putrescentiae* from the allergic subjects were selected for the identification of the major allergenic components by IgE immuno-blotting. The sera from *T. putrescentiae* sensitive subjects were selected for the measurement of IgE responsiveness to allergenic components before and after *D. pteronyssinus* absorption. The frequencies of IgE responsiveness to these allergens were shown as follows: Tyr p 1 (61.3%), Tyr p 2 (79.2%), Tyr p 3 (49.1%), Tyr p 7 (37.7%), Tyr p 8 (44.3%), Tyr p 10 (18.9%) and Tyr p 20 (48.1%), respectively (Figure 4). The major IgE-binding components of *T. putrescentiae* were Group 1 (Tyr p 1) and Group 2 (Tyr p 2) with a molecular weight of 25 kDa (61.3%) and 16 kDa (79.2%). The Group 3 allergen (Tyr p 3) with a molecular weight of 26 kDa (49.1%) was also one of the most prevalent IgE-binding components. The IgE-binding components of Group 8 (Tyr p 8) with a molecular weight of 26 kDa (44.3%) and Group 20 (Tyr p 20) with a molecular weight of 40 kDa (48.1%) also belonged to significant allergens. For further confirmation of cross-reactivity between *D. pteronyssinus*, the *T. putrescentiae*-sensitive sera were selected for performing the IgE immunoblot inhibition with *D. pteronyssinus* absorption. After *D. pteronyssinus* absorption, the IgE-binding frequency of Tyr p 1, Tyr p 2, Tyr p 3, Tyr p 7, Tyr p 8, Tyr p 10 and Tyr p 20 were decreased as 51.9%, 39.6%, 41.5%, 32.1%, 17.9%, 5.7% and 17.9% (Figure 4). The results showed that the IgE-binding frequencies of *T. putrescentiae* allergenic components were decreased, suggesting that there were different levels of cross-reactivity between *T. putrescentiae* and *D. pteronyssinus*. The decreased levels of IgE-binding were higher in Tyr p 8, Tyr p 10 and Tyr p 20 (three-fold decreased), following by Tyr p 2, (two-fold decreased). The level of cross-reactivity between *T. putrescentiae* and *D. pteronyssinus* was trivial in Tyr p 1, Tyr p 3 and Tyr p 7. Results showed the highest level of cross-reactivity between *T. putrescentiae* and *D. pteronyssinus* were Group 10 allergens, the following were Group 20 and Group 8 allergens. The protein sequences of these allergens were compared between *T. putrescentiae* and *D. pteronyssinus* by Basic Local Alignment Search Tool (BLAST), it showed the sequence similarities were over 80% in Group 8, Group 10 and Group 20 allergens (Appendix A). The lower similarities of protein sequences were, respectively in Group 1, Group 3 and Group 7. The analysis of sequence similarity confirmed that it was indeed consistent with the experimental demonstration of cross-reactivity performed by IgE immunoblot inhibition.

### 3.5. The Frequency of IgE-Binding in T. putrescentiae Allergenic Components in the Different Age Group before and after D. pteronyssinus Absorption

The frequencies of IgE-binding to *T. putrescentiae allergenic components* were 44.4% (71/160) in the elderly age group (over or equal 65 years old) and 25.4% (35/138) in the middle-age group (less 40 years old). For further identification of the allergy condition in different age groups, the sera with positive IgE responsiveness to *T. putrescentiae* were selected for the measurement of responsiveness to each *T. putrescentiae* allergenic components before and after *D. pteronyssinus* absorption. Before the *D. pteronyssinus* absorption, the frequencies of IgE-binding to each *T. putrescentiae* allergenic components were shown as follows: Tyr p 1 (63.6%), Tyr p 2 (78.9%), Tyr p 3 (43.7%), Tyr p 7 (39.4%), Tyr p 8 (43.7%), Tyr p 10 (12.7%) and Tyr p 20 (47.9%) in the elderly age group, respectively (Figure 5). In the middle-age group, the frequencies of IgE-binding to allergens were shown as follows: Tyr p 1 (57.1%), Tyr p 2 (80%), Tyr p 3 (60%), Tyr p 7 (34.3%), Tyr p 8 (45.7%), Tyr p 10 (31.4%) and Tyr p 20 (48.6%), respectively (Figure 5). Before *D. pteronyssinus* absorption, the frequencies of IgE-binding to the allergenic components of *T. putrescentiae* in the middle-age group were higher than the elderly age group, except Tyr p 1 and Tyr p 7 (Figure 5).

Before *D. pteronyssinus* absorption, the major allergenic components of *T. putrescentiae* in the middle-age group with age under 40 years old were Tyr p 1 (57.1%), Tyr p 2 (80%) and Tyr p 3 (60%). After the *D. pteronyssinus* absorption, the frequencies of IgE-binding to Tyr p 1 (57.7% vs. 40%), Tyr p 2 (49.3% vs. 20%), Tyr p 7 (35.2% vs. 25.7%) and Tyr p 8 (19.7% vs. 14.3%) in the elderly age group were higher than in middle-age group. The data of statistical analysis showed the frequencies of IgE-binding were significantly reduced after the *D. pteronyssinus* absorption in the Group 10 and Group 8 allergens no matter in the elderly age group or middle-age group (*p* < 0.01; Figure 5). The frequencies of IgE-binding were significantly reduced after the *D. pteronyssinus* absorption in the Group 2 allergen in the middle-age group (*p* < 0.01; Figure 5). There were significant differences between the before and after *D. pteronyssinus* absorption in Group 1 and Group 20 allergens in the middle-age group (*p* < 0.05; Figure 5). Furthermore, an obvious decrease was observed after the *D. pteronyssinus* absorption in Group 2 allergen in the elderly age group (*p* < 0.05; Figure 5). There were higher IgE-binding frequencies of *T. putrescentiae* specific allergens in the elderly age group. These results suggested that majority sensitization of *D. pteronyssinus* was in the middle-age group and the majority sensitization of *T. putrescentiae* was in the elderly age group.

### 3.6. T. putrescentiae Allergy Diagnosis Using Several Combinations of IgE-Responsiveness Prevalence to Allergenic Components

The identification of sensitization to storage mite *T. putrescentiae* is important and essential in clinical diagnosis. Crude extracts based diagnostic methods for identifying mite allergy and sensitization, such as skin prick test, provocation test and extract serology, remain some disadvantages in clinical practice. In this study, we investigated whether *T. putrescentiae* allergenic component combinations of IgE responsiveness can provide more effective and precise methods for clinical diagnosis. Which type of combinations of the IgE responsiveness is the highest precision of prediction accuracy was estimated. The IgE responsiveness to *T. putrescentiae* allergenic components were chosen to composite five combinations, including combination A (Tyr p 1 & Tyr p 2), combination B (Tyr p 2 & Tyr p 3), combination C (Tyr p 3 & Tyr p 7), combination D (Tyr p 1, Tyr p 2 & Tyr p 3) and combination E (Tyr p 2, Tyr p 3 & Tyr p 7) to determine the prediction accuracy. All possible permutations of allergen IgE responsiveness were analyzed to find out the most accurate combination set. Because of the size of the table, only a few combinations of five sets with high predictive value over 80% of *T. putrescentiae* allergy are shown in Table 2. The results showed that the prediction accuracy for *T. putrescentiae* allergy by IgE-responsiveness combination A, B and C could be reached up to eighty percent (84.0–88.7) (Table 2). The prediction accuracy for *T. putrescentiae* allergy by IgE-responsiveness combination D (Tyr p 1, Tyr p 2 & Tyr p 3) and E (Tyr p 2, Tyr p 3 & Tyr p 7) could be achieved to 100% (Table 2). It seems these major allergens are more suitable for diagnostic agents.

The IgE inhibition between *T. putrescentiae* and *D. pteronyssinus* was used to evaluate the cross-reactivity and to identify different components belongs to cross-reacting allergens or species-specific allergens. In order to consider the influence of cross-reactivity from house dust mite, the IgE adsorption experiment was performed. The data of IgE-responsiveness combinations to species-specific allergens with high predictive value over 55% of *T. putrescentiae* allergy are shown in Table 3. After *D. pteronyssinus* absorption, the IgE responsiveness to *T. putrescentiae* allergenic components were chosen to composite seven combinations, including combination A+ (Tyr p 1 & Tyr p 2), combination B+ (Tyr p 2 & Tyr p 3), combination C+ (Tyr p 3 & Tyr p 7), combination D+ (Tyr p 1, Tyr p 2 & Tyr p 3), combination E+ (Tyr p 2, Tyr p 3 & Tyr p 7), combination F+ (Tyr p 2, Tyr p 3, Tyr p 7 &Tyr p 8) and combination G+ (Tyr p 2, Tyr p 3, Tyr p 7, Tyr p 8, Tyr p 10 & Tyr p 20). The prediction accuracy for *T. putrescentiae* allergy by IgE-responsiveness combination D+ (Tyr p 1, Tyr p 2 & Tyr p 3) could be reached up to 90% (Table 3). Moreover, the prediction accuracy of IgE-responsiveness combination H+ (Tyr p 1, Tyr p 2, Tyr p 3, Tyr p 7, Tyr p 8, Tyr p 10 & Tyr p 20) could be achieved to 100%, it showed this combination H+ is more suitable for *T. putrescentiae*-specific diagnosis.

## 4. Discussion

Inhaled allergens are considerable factors in the genesis of allergic diseases. The ubiquitous presence of *T. putrescentiae* in urban and rural settings is an important occupational hazard in agricultural environments or a common allergy-provocation factor in household surroundings [29]. In view of existence abundance and high allergenicity, the house dust mite *D. pteronyssinus*, *D. farinae* and storage mite *T. putrescentiae* are at high levels of infestation and associated with airway hypersensitivity for the development of allergic diseases [30,31]. The prominent appearance and abundance of mites may be influenced by the warm and humid climate in Taiwan. *D. pteronyssinus* is the major cause of allergy in more than 80% of allergic patients with bronchial asthma in Taiwan [8,32], and the major allergens have been identified as Der p 1, Der p 2 and Der p 3 [8]. Most of the allergic patients with mite allergy are almost allergic to both species of mites, *D. pteronyssinus* and *D. farinae*, which are the predominant species and coexist in most geographical regions [33]. The phenomenon can be speculated that it may be cosensitized to *D. pteronyssinus* and *D. farinae* in the household surroundings or possessed high levels of immunologic cross-reactivity between their allergenic components [33]. There is a high prevalence of *T. putrescentiae* sensitization in the general adult population in both Europe [34] and Asia [35]. The prevalence of *T. putrescentiae* sensitization in the United States is also very important that there is 34.3% serum sample sensitized both to *T. putrescentiae* and *D. pteronyssinus* [36]. The storage mite *T. putrescentiae* has often been identified in the house dust samples from South America [37]. *T. putrescentiae* is the most prevalent species of storage mite in the dry-stored food products from house and retail stores in Egypt of Africa [38]. In Spain, where the climate is rainy and temperate, *D. pteronyssinus*-allergic patients also show a very high prevalence (73–83%) of storage mite sensitization [34]. There, 52% of adult individuals over the age of 50 are sensitive to *T. putrescentiae*, which is much higher than *D. pteronyssinus* (22%) in the same age group. Similar findings have been reported in Korea. *T. putrescentiae* is the third most common mite in the indoor environment and around one-third of individuals are allergic to it [35]. Despite the high prevalence of *T. putrescentiae* sensitization in the general adult population, especially in individuals over 50 years old, there are few reports on its clinical significance in elderly populations. It is of interest to evaluate the allergenicity, prevalence of hypersensitivity and clinical relevance of *T. putrescentiae* sensitization in elderly populations, this phenomenon implicates that *D. pteronyssinus* and *T. putrescentiae* may have different sensitization exposure.

Recently, storage mites have been reported to be clinically important allergenic components of dust samples throughout the world, supporting their clinical significance [39]. In our previous report, we demonstrated that 63% of asthmatic patients had concomitant sensitization to storage mite *B. tropicalis* and house dust mite *D. pteronyssinus* by skin tests and the molecular structure and cross-reactivity between Blo t 5 and Der p 5 were estimated at 33% to 43% by IgE inhibition [40]. This suggests that several IgE-binding proteins of storage mites are cross-reactive components, which can be inhibited by *D. pteronyssinus*, but the others are species-specific allergens [40]. In our previous study, we had investigated the relationship between *T. putrescentiae* and *D. pteronyssinus* allergy in allergic rhinitis individuals by using skin prick tests, basophil histamine release assays in vitro and specific IgE measurements [18]. The reactivity between *T. putrescentiae* and *D. pteronyssinus* was investigated and results showed that the major cross-reacting allergen was Group 2 allergens and species-specific allergen was Group 3 allergens [27].

It is also a paramount challenge to develop a small amount of blood for the diagnosis of mite allergy. The study can provide a solution to detect many allergens with a small amount of blood to identify which allergenic components of *T. putrescentiae* to cause immune responses for developing allergic diseases. For allergy diagnosis, the purity and the representativeness of the allergens should also be considered. Since the diagnosis of *T. putrescentiae* allergy is still based on the crude extracts and some of the allergenic components which been found or obtained. Some of these allergenic components possess enzymatic activities causing the instability of allergen preparation which results in IgE-binding activity varied and difficult to make an accurate diagnosis. That is the explanation why the divergences of allergen contents had been mentioned that the levels of both Der p 1 and Der p 2 in *D. pteronyssinus* extract from the different company reveal variability and the proportion variation of Der p 1 and Der p 2 among different extracts are likely to influence their biologic effectiveness [41]. Therefore, the recombinant allergens which reserved immunological activity of the natural allergen can be used for component-resolved-diagnosis (CRD) of the patient’s allergen sensitization profiles of Type I allergy, whereas avoiding disadvantages that are difficult to standardize allergen contents and contain additional undefined nonallergenic components in the crude extracts [42]. The CRD based on recombinant allergens can use to establish IgE sensitization profiles in the polysensitized patients and to realize cosensitization and cross-reactivity of the allergen components [43]. The potential association between allergen components causing allergic symptoms can be further understood using CRD [43]. CRD enables testing for a specific IgE against multiple allergen components, it can be very useful in the diagnosis of anaphylaxis caused by unknown allergens [44]. CRD can also be applied to the diagnosis of food allergy, pollen allergy, occupational asthma, illicit drug hypersensitivity and utility in deciding immunotherapy [44]. The ImmunoCAP^©^ ISAC allergen-microarray (Phadia, Uppsala, Sweden) for the determination of specific IgE against recombinant or purified allergen components has good analytical performance when compared with the traditional method of ImmunoCAP© 250 [45]. In our study, recombinant allergens of Tyr p 2, Tyr p 3, Tyr p 7, Tyr p 8, Tyr p 10 and Tyr p 20 were cloned and generated to detect specific IgE in the serum of *T. putrescentiae* sensitized patients. The prediction accuracy for *T. putrescentiae* allergy by IgE-responsiveness combination D (Tyr p 1, Tyr p 2 & Tyr p 3) and E (Tyr p 2, Tyr p 3 & Tyr p 7) could be achieved to 100%. It seems the prediction combinations of these recombinant allergens from major allergens are suitable for diagnostic agents. Recombinant allergens have provided us with high accuracy and efficiency that can improve the diagnosis of allergy.

Childhood asthma and adult-onset asthma are a globally significant immune-disorder disease with major public health consequences known to share many of the same causes and triggers [46]. Allergic sensitizations and environmental exposures including mite allergens are associated with asthma severity in children and adolescents [46]. The component-specific IgE responses associated with diverse risk of asthma existence, continuance and severity have been identified using component-resolved diagnostics rather than standard skin tests and blood tests to the whole allergen extracts in children [47]. The evolution of IgE responses to multiple allergen components throughout allergy pathogenesis described in this study, which may facilitate the development of better diagnostic and prognostic biomarkers for allergic diseases.

It is feasible to generate a panel of *T. putrescentiae* allergenic components and develop an immunoassay kit to identify the IgE-mediated hypersensitivity of *T. putrescentiae*. A patent for the diagnostic kit (patent No. I 467177) for *T. putrescentiae* allergy including the allergens of rTyr p2, rTyr p3, rTyr p 7, rTyr p 8, rTyr p 10 and rTyr p 20 has been registered and certified in Taiwan, the Republic of China (expired time 04/22/2032). This is based on a combination with high-frequency IgE-binding component of Tyr p 2 and the high species-specific allergenic components of Tyr p 3 & Tyr p 7, a refined microarray panel of allergens. The development of diagnostic kits can be identified *T. putrescentiae* IgE-mediated hypersensitivity more specifically the detectable allergens would broaden involved more species-specific allergens in *T. putrescentiae*, which avoid the cross-reactive allergen of *D. pteronyssinus* whenever possible.

*Tyrophagus putrescentiae* is one of the most common storage mites found in the storage rooms, kitchen and frequently contaminated in the mushrooms, farina and pet food [48]. There are more than 80% of dust samples contained mites, including house dust mites (*D. pteronyssinus* and *D. farinae*) and storage mites (*T. putrescentiae*) be identified in the dust samples [40,49]. Antigens of storage mite are predominantly found in the kitchens and antigens of house dust mite are mainly detected in the bedrooms, especially the *T. putrescentiae* is the most prevalent mite in all room types [49]. The mite bodies and feces of *T. putrescentiae* can be accumulated in the contaminated foods or residential surroundings causing food allergy and airway hypersensitivity [48,50]. Although there are at least 36 allergenic components reported in *D. pteronyssinus*, in contrast only a few allergens of *T. putrescentiae* been identified and characterized. In the currently identified allergens of *T. putrescentiae*, there are some major allergens such as Tyr p 1, Tyr p 2 and Tyr p 3 with higher than 50% IgE-binding activity in the *T. putrescentiae* sensitized patients caused hypersensitivities.

The seven allergenic components were used to identify the IgE-binding activity and to confirm the IgE immunoblot inhibition, it showed that all of the IgE-binding activities decreased after *D. pteronyssinus* absorption indicating certain levels of cross-reactivity between *T. putrescentiae* and *D. pteronyssinus*. In this study, the highest level of cross-reactivity allergen between *T. putrescentiae* and *D. pteronyssinus* was the Group 10 mite allergen. The Group 10 mite allergens are involved in muscle contraction in invertebrates and processed high levels of cross-reactivity between mites, shrimp and insects in seafood-allergic patients, referred to as pan-allergens [51]. The cross-reactivity between *T. putrescentiae* and *D. pteronyssinus* was confirmed after the IgE absorption and protein sequence comparison; it showed the most cross-reactive allergens were Group 10, Group 20 and Group 8. Another highly conserved component had been identified as Group 2 allergen, similar findings demonstrated that the IgE-binding activity of Tyr p 2 could be extremely absorbed by Der p 2 and the Group 2 allergens are the major cross-reactive allergen of *T. putrescentiae* and *D. pteronyssinus* [18]. The Der p 2 is a lipid-binding protein [52], which binds lipopolysaccharides (LPS) in a style similar to binding by myeloid differentiation protein-2 (MD-2) onto Toll-like receptor-4 (TLR-4) [53]. The allergenicity of Group 2 mite allergens may be heightened by a similar structure interaction [54], therefore about 80–100% of mite-allergic patients are primarily sensitized to Der p 2 [8], the results as our research found that about 80% were sensitized to Tyr p 2 in the *T. putrescentiae*-allergic patients. Most the surface residues are conserved leading to similar epitopes forming the structural basis of cross-reactivity between Group 2 allergens of dust mites [55]. Not all of the allergenic components possessed high levels of cross-reactivity between *T. putrescentiae* and *D. pteronyssinus*, such as Tyr p 3 and Tyr p 7, the reduction of IgE-binding activity after *D. pteronyssinus* absorption were trivial. The similar finding showed there are poor correlations of serum specific IgE to Tyr p 3 and Der p 3, which with poor cross-reactivity between them and the inhibition degree by Group 3 allergens (Tyr p 3) is lower than by Group 2 allergens (Tyr p 2) in the *T. putrescentiae*-allergic patients [18]. Overall, it suggests Tyr p 3 is a major non-cross-reactive component belongs to a *T. putrescentiae* specific allergen and may serve as a suitable candidate for a differential diagnostic marker for *T. putrescentiae* hypersensitivity.

The main limitations of this study are related to the usage of allergen groups and the recruitment of *T. putrescentiae*-allergic patients. Not all of the identified allergens of *T. putrescentiae* were used for the development of diagnostic kits in this study. Although not all allergenic components are included, most major allergens are included for diagnosis and the prediction accuracy is close to 100%. In this study, just 106 *T. putrescentiae*-allergic patients were used to check the accuracy of diagnostic kits. Actually, a total of 298 allergic patients were recruited in the Clinic of the Division of Allergy, only about 1/3 of allergic patients were allergic to *T. putrescentiae* with IgE positive responses. This is can be contributed to more than 80% of the dust samples contained house dust mites including *D. pteronyssinus* and *D. farinae* in Taiwan [56]. Most of these allergic patients were exposed to them by contact with mattresses or carpets in the household environment then sensitized to house dust mites of *D. pteronyssinus* and *D. farinae* [57]. Because of their abundance and high allergenicity of *D. pteronyssinus* and *D. farinae*, the major predominant mites that elicit positive IgE responses and provoke airway hypersensitivity in Taiwan [48]. Another limitation was that we do not evaluate the associations between the levels of other antibodies except the IgE and severity of the allergy. The pathogenesis and severities of allergic diseases are actually related to many factors such as inflammatory mediators (cytokines or chemokines), granulocytes (mast cells, basophils and eosinophils), T cell differentiation and airway remodeling [58]. In this study, we focused more on the IgE-mediated hypersensitivity and improved the accuracy of the diagnostic combination of allergenic components.

## 5. Conclusions

The storage mite *T. putrescentiae* is an important occupational hazard in agricultural environments, accident ingestion of mite-contaminated foods and ubiquitous presence in urban or rural settings, it is also a common allergy-provocation factor in household surroundings. So, the diagnosis method of *T. putrescentiae* allergy with convenience and accuracy is necessary for clinical treatment. The allergy diagnosis based on the crude extracts still contains some disadvantages and insufficiencies. The recombinant allergens can be used for understanding the sensitization profiles of allergic patients. In this study, we found out prediction accuracy of IgE responsiveness for *T. putrescentiae* with allergen combination (Tyr p 1, Tyr p 2 and Tyr p 3) could be achieved to high precision (100%). Avoiding the allergen cross-reactivity from predominant mite *D. pteronyssinus* performed by IgE absorption, the allergenic components of Tyr p 3 and Tyr p 7 have been identified as species-specific allergens. The IgE responsiveness to species-specific allergens is more suitable for the diagnosis of *T. putrescentiae* allergy focusing on IgE-mediated hypersensitivity.

## Figures and Tables

**Figure 1 diagnostics-10-00665-f001:**
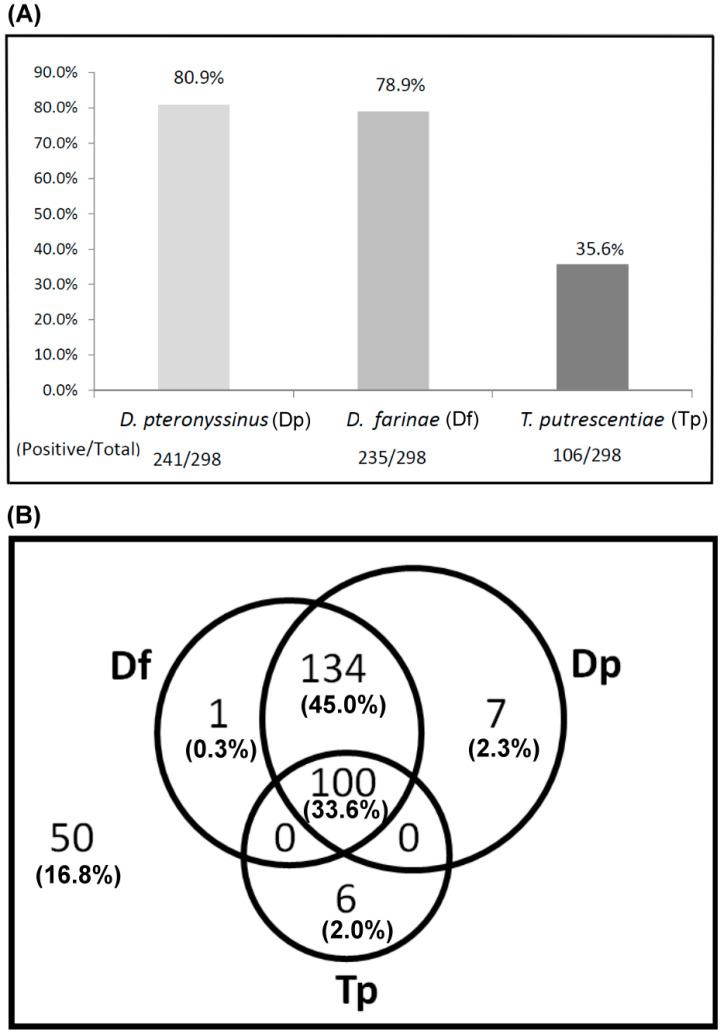
Prevalence of sensitization to *D. pteronyssinus*, *D. farinae* and *T. putrescentiae* of allergic patients were determined by ELISA. A total of 298 allergic patients were recruited (n = 298). (**A**) Positive rate of IgE response was presented as a bar histogram; (**B**) cosensitization to these mites was presented as a Venn diagram; the overlapping parts meant multiple sensitizations to these mites. Outside the circle indicated without IgE response against these species of mites. Dp—*D. pteronyssinus*; Df—*D. farinae*; Tp—*T. putrescentiae.*

**Figure 2 diagnostics-10-00665-f002:**
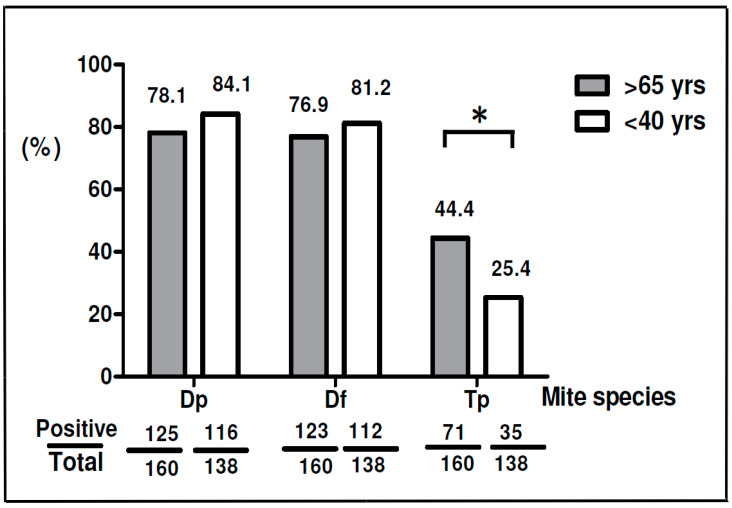
Prevalence of sensitization to *D. pteronyssinus*, *D. farinae* and *T. putrescentiae* in different age groups. Gray bars represent patients over 65 years old; white bars represent patients under 40 years old. * *p* < 0.05, the *p*-values less than 0.05 were considered statistically significant.

**Figure 3 diagnostics-10-00665-f003:**
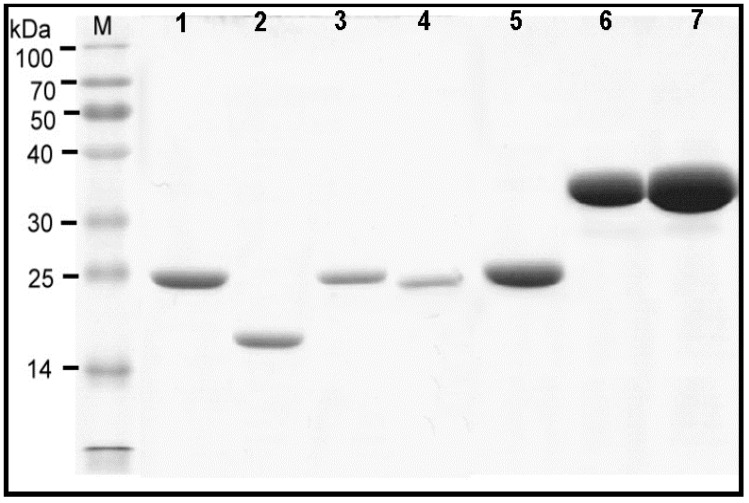
Protein profiles of *T. putrescentiae* recombinant allergens. Seven allergens were generated and presented on 12% SDS-PAGE. Lane 1: rTyr p 1, approximately at 25 kDa. Lane 2: rTyr p 2 at around 16 kDa. Lane 3 to lane 5: rTyr p 3, rTyr p 7 and rTyr p 8, at around 24–26 kDa. Lane 6 and lane 7: rTyr p 10 and rTyr p 20, at around 37 to 40 kDa. M—protein marker.

**Figure 4 diagnostics-10-00665-f004:**
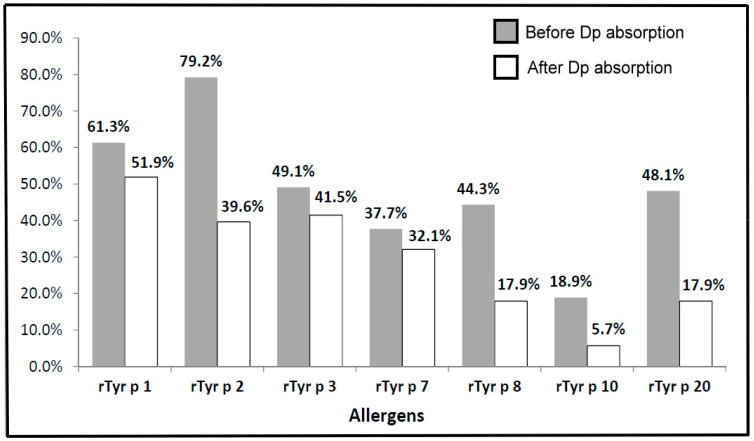
Frequencies of IgE responsiveness to rTyr p 1, rTyr p 2, rTyr p 3, rTyr p 7, rTyr p 8, rTyr p 10 and rTyr p 20. Gray bars represent frequencies of IgE responsiveness before *D. pteronyssinus* absorption; white bars represent frequencies of IgE responsiveness after *D. pteronyssinus* absorption.

**Figure 5 diagnostics-10-00665-f005:**
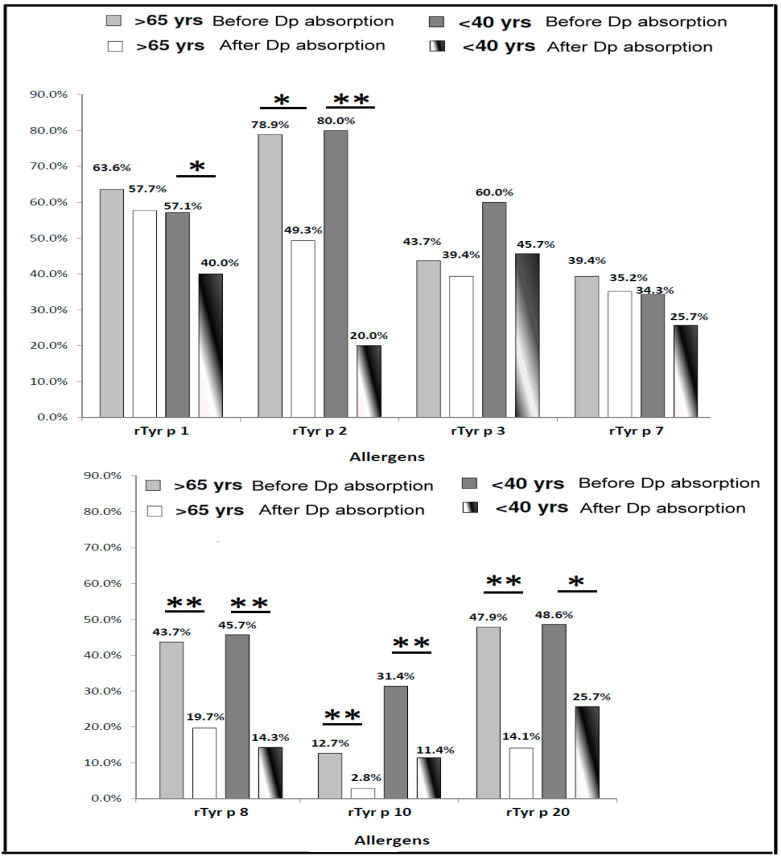
Prevalence of IgE responsiveness to *Tyrophagus putrescentiae* allergenic components in different age group. Light gray bars represent frequencies of IgE responsiveness before *D. pteronyssinus* absorption; white bars represent frequencies of IgE responsiveness after *D. pteronyssinus* absorption of patients over 65 years old (total n = 71); dark gray bars represent frequencies of IgE responsiveness before *D. pteronyssinus* absorption; white-black bars represent frequencies of IgE responsiveness after *D. pteronyssinus* absorption of patients under 40 years old (total n = 35). * for *p* < 0.05 when compared with before *D. pteronyssinus* absorption; ** for *p* < 0.01 when compared with before *D. pteronyssinus* absorption.

**Table 1 diagnostics-10-00665-t001:** *Tyrophagus putrescentiae* Allergens in the GenBank of National Center for Biotechnology Information.

Allergen Name	Protein Family or Biologic Function	MW cDNA ^1^ (SDS-PAGE) (KDa)	GenBank Accession No.
Tyr p 1	Cysteine protease	25	ABM53753
Tyr p 2	Niemann-Pick C2 (NPC2)	16	ABU97477
Tyr p 3	Trypsin-like serine protease	26	ABZ81991
Tyr p 7	Bactericidal permeability-increasing like protein	25	ABM53750
Tyr p 8	Glutathione S transferases	26	AGG10560
Tyr p 10	Tropomyosin	37	AAT40866
Tyr p 20	Arginine kinases	40	MT900252

^1^ MW calculated from cDNA and referenced the allergen on SDS-PAGE.

**Table 2 diagnostics-10-00665-t002:** Diagnostic combination for *Tyrophagus putrescentiae* allergy.

Prevalence of IgE Responsiveness to *T. putrescentiae* Allergens							
Allergens	Tyr p 1	Tyr p 2	Tyr p 3	Tyr p 7	Tyr p 8	Tyr p 10	Tyr p 20
IgE positive	61.3%	79.2%	49.1%	37.7%	44.3%	18.9%	48.1%
Combination A (Tyr p 1 & 2)	88.7%						
Combination B (Tyr p 2 & 3)		84.0%					
Combination C (Tyr p 3 & 7)			86.0%				
Combination D (Tyr p 1, 2, 3)	100%						
Combination E (Tyr p 2, 3, 7)		100%					

Total of 106 patients with IgE responsiveness to *Tyrophagus putrescentiae.*

**Table 3 diagnostics-10-00665-t003:** Diagnostic combination for *T. putrescentiae* allergy after Dp absorption.

Prevalence of IgE Responsiveness to *T. putrescentiae* Allergens Exclude Dp Cross-Reaction								
Allergens	Tyr p 1	Tyr p 2	Tyr p 3		Tyr p 7	Tyr p 8	Tyr p 10	Tyr p 20
IgE positive	51.9%	39.6%	41.5%		32.1%	17.9%	5.7%	17.9%
Combination A+ (Tyr p 1 & 2)	78.3%							
Combination B+ (Tyr p 2 & 3)		67.9%						
Combination C+ (Tyr p 3 & 7)				56.6%				
Combination D+ (Tyr p 1,2, 3)	90.1%							
Combination E+ (Tyr p 2, 3, 7)		79.2%						
Combination F+ (Tyr p 2, 3, 7, 8)		87.0%						
Combination G+ (Tyr p 2, 3, 7, 8, 10, 20)		88.0%						
Combination H+ (Tyr p 1, 2, 3, 7, 8, 10, 20)	100%							

Total of 106 patients with IgE responsiveness to *Tyrophagus putrescentiae.*

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
