# Peer review of "Identification the Cross-Reactive or Species-Specific Allergens of Tyrophagus putrescentiae and Development Molecular Diagnostic Kits for Allergic Diseases"

_diagnostics, 2020, doi:10.3390/diagnostics10090665_

Round 1

Reviewer 1 Report

The manuscript describes the production of recombinant allergens from storage mites and their potential use as differential diagnostic markers for allergen species sensitization. Cross-reactivity among allergens was also evaluated. The manuscript presents an interesting idea with already required patenting, which seems to be well designed, although not always well presented (results). The manuscript could be considered for publication on the Diagnostics journal after performing the recommended alterations.

Comments

The English language needs to be fully revised. Along the manuscript there are several sentences that are not well constructed (probably missing words, prepositions, etc), which hampers the clarity of the paper. Please revise carefully.

Revise references, they are not well formatted. They must follow the established guidelines for authors.

Line 52 - Immunoglobulins are by definition antibodies, therefore IgE antibody is redundant. Please delete.

line 56 - The authors could highlight here the allergens that have been registered at the WHO/IUIS.

Line 86-87 - The sentence is not well constructed. Please rephase.

Figure 2 - Did the authors evaluate the statistical differences for mite species sensitization within each group (adults versus elderly)? Please add analysis.

Lines 168-170 - Were these results expect? How do the authors justify those strong decreases in IgE-binding frequencies, especially for Tyr p 2? Has this fact been identified by other authors?

Lines 187-193 - Did the authors find any statistical difference is the IgE-binding frequencies among the groups? Please add statistical analysis.

Table 2 - Tyr p 2 is one of the allergens presenting the highest differences before and after Dp absorption. Any particular reason for these results?? Please explain.

Table 2 - Format Table 2.

Both Table 3 must be combined in one with clear identification of the prediction accuracy before and after Dp absorption.

Table 3 - there is no correspondence among the combinations used to fill both tables 3. This is very confusing. For instance, combination C is Tyr p 1,2,3 in the first table 3 and in the second is combination D (Tyr p 1,2,3). This makes no sense, please harmonize the combinations in table!!!!

Line 230-240 - Please explain why there were only 5 combinations to determine prediction accuracy before Dp absorption and there are 7 after?

Lines 238-240 - It seems clear to me if the authors include practically all allergens, the prediction efficacy will be 100%. What the authors do not comment is the prediction accuracy before and after Dp absorption. Relating the data from both table 3, I would recommend for the combination C+ (Tyr p 1, 2, 3), since it presents minor differences in prediction accuracy compared to D. Please comment on that.

Line 257 - the references used have already several years. Are there any more recent prevalence studies that could be cited here, thus reflecting the current situation in Europe and Asia? How about Americas and Africa, are there any relevant data on storage mite prevalence? Please add information or comment on the lack of evidence.

Line 270 - Change to "... had concomitant sensitization..."

Lines 299-301 - Are there more recent findings relating CRD and the evaluation of crude extracts or recombinant proteins? Please comment on the recent advances.

Line 307 - the correct term is sensitized patient, not sensitive. Analytical methods can be sensitive, but patients are sensitized. Please correct.

Line 308 - In lines 238-240 the authors say "...prediction accuracy of IgE responsiveness combination G+ (Tyr p 1, 2, 3, 7, 8, 10 & 20) could be achieved to 100%, it showed this combination G+ is more suitable for T. putrescentiae-specific diagnosis." This is confusing because authors are not using a systematic nomenclature for the combination of allergenic proteins to assess the IgE responsiveness to T. putrescentiae. Please clarify after using harmonized combination nomenclature.

line 358 - music contraction?? Do you mean muscle contraction??

Line 375 - Change to "...suitable candidate for a differential diagnostic marker…"

Author Response

Reply for Review-1

Comments and Suggestions for Authors

The manuscript describes the production of recombinant allergens from storage mites and their potential use as differential diagnostic markers for allergen species sensitization. Cross-reactivity among allergens was also evaluated. The manuscript presents an interesting idea with already required patenting, which seems to be well designed, although not always well presented (results). The manuscript could be considered for publication on the Diagnostics journal after performing the recommended alterations.

Comments

Q: The English language needs to be fully revised. Along the manuscript there are several sentences that are not well constructed (probably missing words, prepositions, etc), which hampers the clarity of the paper. Please revise carefully. Revise references, they are not well formatted. They must follow the established guidelines for authors.

Reply: We acknowledge your thoughtful insights and comments. Your suggestions make this article better. The entire article and reply comments have been requested by professional grammar experts to make amendments. More appropriate description and more recent references have been added in the revised manuscript.

Q: Line 52 - Immunoglobulins are by definition antibodies, therefore IgE antibody is redundant. Please delete.

Reply: Thank you for pointing out this typo. The “…antibody…” has been deleted as follows. (line 52)

  1. pteronyssinus occurs at high levels of infestation in houses around the world and with high IgE frequency, which the mite extracts have indicated over 36 different allergenic components can induce IgE in patients and about 32 allergens been identified (9).

Q: line 56 - The authors could highlight here the allergens that have been registered at the WHO/IUIS.

Reply: We thank you very much for your suggestion. The allergens of T. putrescentiae registered at the WHO/IUIS have been highlighted as follows. (lines 59-62)

According to the information of World Health Organization/International Union of Immunological Societies (WHO/IUIS) Allergen Nomenclature Sub-Committee (http://www.allergen.org/), a total of 9 allergens of T. putrescentiae have been registered including Tyr p 2, Tyr p 3, Tyr p 8, Tyr p10, Tyr p 28, Tyr p 34, Tyr p 35 and Tyr p 36 (12).

Q: Line 86-87 - The sentence is not well constructed. Please rephase.

Reply: The more appropriate description of this sentence has been rephased as follows. (lines 90-92)

So far, no relevant researches focusing on the development of differential diagnostic kit for T. putrescentiae allergy have been investigated.

Q: Figure 2 - Did the authors evaluate the statistical differences for mite species sensitization within each group (adults versus elderly)? Please add analysis.

Reply: We acknowledge your thoughtful insights and comments. The more detail description of analysis differences have added as follows. (lines 125-130)

Additionally, the analysis data showed there were 44.4% (71/160) elderly subjects sensitive to T. putrescentiae while there were only 25.4% (35/138) young adult subjects allergic to T. putrescentiae. The prevalence of sensitization to D. pteronyssinus or D. farinae was slightly higher in the young adult subjects, while the prevalence of sensitization to T. putrescentiae was significantly higher in elderly subjects than in young adult subjects (44.4% vs. 25.4%)(p<0.05).

Q: Lines 168-170 - Were these results expect? How do the authors justify those strong decreases in IgE-binding frequencies, especially for Tyr p 2? Has this fact been identified by other authors?

Reply: Yes, the results were as we expected and similar to the previous study. The highlighted finding and explanation have added in the Result and Discussion Section as follows.

In the Result section: (lines 177-179)

The decreased level of IgE-binding was highest in Tyr p 10 (three-fold decreased), following by Tyr p 2, Tyr p 8, Tyr p 20 (two-fold decreased).

In the Discussion section: (lines 314-316)

The reactivity between T. putrescentiae and D. pteronyssinus was investigated and results showed that the major cross-reacting allergen was group 2 allergens and species-specific allergen was group 3 allergens (39).

In the Discussion section: (lines 386-389)

Another highly conserved component had been identified as group 2 allergen, similar findings demonstrated that the IgE-binding activity of Tyr p 2 could be extremely absorbed by Der p 2 and the group 2 allergens are the major cross-reactive allergen of T. putrescentiae and D. pteronyssinus (18).

In the Discussion section: (lines 391-394)

The allergenicity of group 2 mite allergens might be heightened by a similar structure interaction (53), therefore about 80-100% of mite-allergic patients are primarily sensitized to Der p 2 (8), the results as our research found that about 80% were sensitized to Tyr p 2 in the T. putrescentiae allergic patients.

In the Discussion section: (lines 394-396)

The majority of the surface residues are conserved leading to similar epitopes forming the structural basis of cross-reactivity between group 2 allergens of dust mites (54).

Q: Lines 187-193 - Did the authors find any statistical difference is the IgE-binding frequencies among the groups? Please add statistical analysis.

Reply: We appreciated your suggestion. The statistical analysis has been added in the Result section and Figure 5 as follows. (lines 222-229)

The data of statistical analysis showed the frequencies of IgE-binding were significantly reduced after the D. pteronyssinus absorption in the group 10 and group 8 allergens no matter in the elder-age group or mid-age group (p<0.01; Figure 5). The frequencies of IgE-binding were significantly reduced after the D. pteronyssinus absorption in the group 2 allergen in the mid-age group (p<0.01; Figure 5). There were significant differences between the before and after D. pteronyssinus absorption in group 1 and group 20 allergens in the mid-age group (p<0.05; Figure 5). Furthermore, an obvious decrease was observed after the D. pteronyssinus absorption in group 2 allergen in the elder-age group (p<0.05; Figure 5).

Q:  Table 2 - Tyr p 2 is one of the allergens presenting the highest differences before and after Dp absorption. Any particular reason for these results?? Please explain.

Reply: We thank you for your comment. The particular reasons for these results, references and explanation have been supplemented in the Discussion section as follows.

Another highly conserved component had been identified as group 2 allergen, similar findings demonstrated that the IgE-binding activity of Tyr p 2 could be extremely absorbed by Der p 2 and the group 2 allergens are the major cross-reactive allergen of T. putrescentiae and D. pteronyssinus (18). (lines 386-389)

The allergenicity of group 2 mite allergens might be heightened by a similar structure interaction (53), therefore about 80-100% of mite-allergic patients are primarily sensitized to Der p 2 (8), the results as our research found that about 80% were sensitized to Tyr p 2 in the T. putrescentiae allergic patients. (lines 391-394)

The majority of the surface residues are conserved leading to similar epitopes forming the structural basis of cross-reactivity between group 2 allergens of dust mites (54). (lines 394-396)

Q: Table 2 - Format Table 2.

Reply: We appreciate your valuable suggestion. The original Table 2 has been replaced as Figure 5. This presentation is indeed clearer (Line 209).

Q: Both Table 3 must be combined in one with clear identification of the prediction accuracy before and after Dp absorption.

Reply: We thank you for your comment. In order to express the experimental results more clearly, the original Table 3 was divided into two tables as Table 2 and Table 3. 

Q: Table 3 - there is no correspondence among the combinations used to fill both tables 3. This is very confusing. For instance, combination C is Tyr p 1,2,3 in the first table 3 and in the second is combination D (Tyr p 1,2,3). This makes no sense, please harmonize the combinations in table!!!!

Reply: We thank you for the constructive criticism. We found this inappropriate layout indeed confused readers. In the original manuscript, the combination of prediction accuracy more than 60% was presented. The Combination C+ (Tyr p 3 & Tyr p 7) of prevalence of IgE responsiveness to T. putrescentiae after the D. pteronyssinus absorption has been added in the revised Table 3. The combination of allergens will be consistent in the revised Table 2 and Table 3. The combination C or C+ meant Tyr p 3 & Tyr p 7. The combination D or D+ meant Tyr p 1, Tyr p 2 and Tyr p 3. The description has been corrected as follows. (lines 267-271)

The prediction accuracy for T. putrescentiae allergy by IgE responsiveness combination D+ (Tyr p 1, Tyr p 2 & Tyr p 3) could be reached up to 90% (Table 3). And the prediction accuracy of IgE responsiveness combination H+ (Tyr p 1, Tyr p 2, Tyr p 3, Tyr p 7, Tyr p 8, Tyr p 10 & Tyr p 20) could be achieved to 100%, it showed this combination H+ is more suitable for T. putrescentiae-specific diagnosis.

Q: Line 230-240 - Please explain why there were only 5 combinations to determine prediction accuracy before Dp absorption and there are 7 after?

Reply: We are very grateful for your valuable comments. Supplementary instructions list as follows. (lines 246-248)

All possible permutations of allergen IgE responsiveness were analyzed to find out the most accurate combination set. Because of the size of the table, only a few combinations of five sets with high predictive value over 80% of T. putrescentiae allergy were shown in Table 2.

In order to consider the influence of cross-reactivity from house dust mite, the IgE adsorption experiment was performed. The data of IgE responsiveness combinations to species-specific allergens with high predictive value over 55% of T. putrescentiae allergy were shown in the Table 3. (lines 259-262)

Q: Lines 238-240 - It seems clear to me if the authors include practically all allergens, the prediction efficacy will be 100%. What the authors do not comment is the prediction accuracy before and after Dp absorption. Relating the data from both table 3, I would recommend for the combination C+ (Tyr p 1, 2, 3), since it presents minor differences in prediction accuracy compared to D. Please comment on that.

Reply: We thank you for the suggestion. We found this inappropriate layout indeed confused readers. In the original manuscript, the combination of prediction accuracy more than 60% was presented. The description has been corrected as follows. (lines 264-268)

The Combination C+ (Tyr p 3 & Tyr p 7) of prevalence of IgE responsiveness to T. putrescentiae after the D. pteronyssinus absorption has been added in the revised Table 3. The combination of allergens will be consistent in the revised Table 2 and Table 3. The combination C or C+ meant Tyr p 3 & Tyr p 7. The combination D or D+ meant Tyr p 1, Tyr p 2 and Tyr p 3.

Q: Line 257 - the references used have already several years. Are there any more recent prevalence studies that could be cited here, thus reflecting the current situation in Europe and Asia? How about Americas and Africa, are there any relevant data on storage mite prevalence? Please add information or comment on the lack of evidence.

Reply: We acknowledge your thoughtful insights and comments. The relevant information on storage mite prevalence has added as follows in the Discussion section. (lines 290-294)

The prevalence of T. putrescentiae sensitization in the United States is also very important that there is 34.3% serum sample sensitized both to T. putrescentiae and D. pteronyssinus (34). The storage mite- T. putrescentiae has often been identified in the house dust samples from South America(35). T. putrescentiae is the most prevalent species of storage mite in the dry-stored food products from house and retail stores in Egypt of Africa (36).

Q: Line 270 - Change to "... had concomitant sensitization..."

Reply: The sentence has been changed to “had concomitant sensitization” as follows. (line 307)

In our previous report, we demonstrated that 63% of asthmatic patients had concomitant sensitization to storage mite B. tropicalis and house dust mite D. pteronyssinus by skin tests, and the molecular structure and cross-reactivity between Blo t 5 and Der p 5 were estimated at 33% to 43% by IgE inhibition (38).

Q: Lines 299-301 - Are there more recent findings relating CRD and the evaluation of crude extracts or recombinant proteins? Please comment on the recent advances.

Reply: We appreciate your comments. The more detail descriptions about recent findings relating CRD have been added in the Discussion section and reference list as follows. (lines 332-338)

The CRD based on recombinant allergens can use to establish IgE sensitization profiles in the polysensitized patients and to realize co-sensitization and cross-reactivity of the allergen components (42). The potential association between allergen components causing allergic symptoms can be further understood using CRD (42). CRD enables testing for a specific IgE against multiple allergen components, it can be very useful in the diagnosis of anaphylaxis caused by unknown allergens (43). CRD can also be applied to the diagnosis of food allergy, pollen allergy, occupational asthma, illicit drug hypersensitivity, and utility in deciding immunotherapy (43).

Q: Line 307 - the correct term is sensitized patient, not sensitive. Analytical methods can be sensitive, but patients are sensitized. Please correct.

Reply: As per your suggestion, we have corrected “sensitive patients” as “sensitized patients” with an underline. (line 343)

Q: Line 308 - In lines 238-240 the authors say "...prediction accuracy of IgE responsiveness combination G+ (Tyr p 1, 2, 3, 7, 8, 10 & 20) could be achieved to 100%, it showed this combination G+ is more suitable for T. putrescentiae-specific diagnosis." This is confusing because authors are not using a systematic nomenclature for the combination of allergenic proteins to assess the IgE responsiveness to T. putrescentiae. Please clarify after using harmonized combination nomenclature.

Reply: We appreciated your valuable and useful suggestion. The improper nomenclatures have been revised in the Result section. (lines 268-271)

And the prediction accuracy of IgE responsiveness combination H+ (Tyr p 1, Tyr p 2, Tyr p 3, Tyr p 7, Tyr p 8, Tyr p 10 & Tyr p 20) could be achieved to 100%, it showed this combination H+ is more suitable for T. putrescentiae-specific diagnosis.

Q: line 358 - music contraction?? Do you mean muscle contraction??

Reply: Thank you for pointing out this typo. The sentence has been corrected as “muscle” contraction (line 384).

The group 10 mite allergens are involved in muscle contraction in invertebrates and processed high levels of cross-reactivity between mites, shrimp, and insects in seafood-allergic patients, referred to as pan-allergens (50).

Q: Line 375 - Change to "...suitable candidate for a differential diagnostic marker…"

Reply: We appreciate for your suggestion. The sentence has been changed as follows in the Discussion section. (line 403)

Overall, it suggests Tyr p 3 is a major non-cross reactive component belongs to a T. putrescentiae specific allergen and may serve as a suitable candidate for a differential diagnostic marker for T. putrescentiae hypersensitivity.

Reviewer 2 Report

Brief summary:

The aimed of this study was to investigate the IgE responsiveness profiles of T. putrescentiae allergic patients and to develop a molecular diagnostic method which allows a specific diagnosis of this individuals. Storage mite allergen components had been characterized as cross-reacting or specie-specific allergens and effective combinations of recombinant allergens for clinical diagnosis have been suggested. Finally, authors propose an allergen combination suitable for T. putrescentiae allergy specific diagnosis avoiding the cross-reactivity of D. pteronyssinus.

General comments:

After careful reading of the manuscript, this reviewer can say that the authors have made a great and very interesting work. In any case, this reviewer considers that there are some aspects that could be improved.

On the one hand, when combination of different allergens are analyzed to determine the better one as a T. putrescentiae diagnostic method authors only check the sensitivity of the selected allergens. Were their specificity verified? Did the authors check the prediction accuracy using samples from individuals allergic to D. pteronyssinus, but not to T. putrescentiae?

On the other hand, regarding the statistics, authors should indicate if they use parametric or non-parametric test in their analysis. They should check the distribution of their samples before choosing the one or the other.

Finally, this reviewer has detected some format errors in the references. It should be checked.

Author Response

Reviewer-2

Brief summary:

The aimed of this study was to investigate the IgE responsiveness profiles of T. putrescentiaeallergic patients and to develop a molecular diagnostic method which allows a specific diagnosis of this individuals. Storage mite allergen components had been characterized as cross-reacting or specie-specific allergens and effective combinations of recombinant allergens for clinical diagnosis have been suggested. Finally, authors propose an allergen combination suitable for T. putrescentiae allergy specific diagnosis avoiding the cross-reactivity of D. pteronyssinus.

General comments:

Q: After careful reading of the manuscript, this reviewer can say that the authors have made a great and very interesting work. In any case, this reviewer considers that there are some aspects that could be improved.

Reply: We appreciate the fact that you find our work interesting. We have now modified the text as per your suggestions.

Q: On the one hand, when combination of different allergens are analyzed to determine the better one as a T. putrescentiae diagnostic method authors only check the sensitivity of the selected allergens. Were their specificity verified? Did the authors check the prediction accuracy using samples from individuals allergic to D. pteronyssinus, but not to T. putrescentiae

Reply: Thank you for reviewing my article very carefully and give us valuable suggestion. This is a very good experimental strategy that we never thought of. In this study, we may not have enough serum approved by the Institutional Review Board of TCVGH to validate this experiment. We need to recruit other allergic patients for further study to verify the prediction accuracy to D. pteronyssinus in the future.

Q: On the other hand, regarding the statistics, authors should indicate if they use parametric or non-parametric test in their analysis. They should check the distribution of their samples before choosing the one or the other.

Reply: We appreciated your suggestions. The more detail description about the Statistical analysis has been added in the “Materials and Methods” Section. (lines 529-531)

Differences between the mid-age and elder-age population of the T. putrescentiae allergy were analyzed by the non-parametric alternative to the unpaired two-sample t-test. The p-values of differences less than 0.05 were considered statistically significant.

Q: Finally, this reviewer has detected some format errors in the references. It should be checked.

Reply: We acknowledge your thoughtful insights and comments. Your suggestions make this article better. More appropriate description and more recent references have been added in the revised manuscript.

Reviewer 3 Report

The paper, “Identification the cross-reactive or species-specific allergens of Tyrophagus putrescentiae and development molecular diagnostic kits for allergic diseases”, describes studies of new and previously identified allergens from T. putrescentiae (Tp) and their applicability to a diagnostic kit.  The paper contains valuable allergological information about mite allergens and cross reactivity with house dust mites.  The Tp species is not as well studied as house dust mites and the paper represents good new information. There are a few major and minor considerations that should be addressed prior to publication.

Major Scientific

A quick look at www.allergen.org reveals that several of these allergens have not been officially named by the WHO/IUIS allergen nomenclature committee.  This must be done prior to acceptance for publication.  Instructions for getting an officially sanctioned name for Tyr p 1, Tyr p 7, and Tyr p 20 can be found at the website and usually takes less than 2 weeks.  This helps keep names correct in the scientific literature and would be a significant contribution to science.  To repeat, this has to be done.

Major Stylistic

The discussion is far too long and could potentially be cut in half.  For example, the paragraphs that begin on line 280 (blood sampling), and 321 (COPD) and not totally applicable or could be cut significantly. The next to last paragraph on the limitations of the study and the conclusion paragraph are the most salient to remain.

The authors should utilize a English language professional to correct numerous typos and somewhat silly errors like line 358, “The group 10 mite allergens are involved in music contraction in invertebrates… ” or the nonsensical term in line 99 “IgE fullness”.

Minor points

Combination D and G+ are named but not explained in the abstract so do not provide valuable information. Please revise to explain succinctly.

One is left wondering about the study population in total.  Please provide more demographic information, preferably in tabular format, about the age ranges, severity of diseases, diagnoses etc that might be relevant.  For example, it is confusing in Figure 2, were there no patients 41-64?

Line 160.  The definition of major allergen is usually prevalence of >50%.  So technically Tyr p 3 is not a major allergen in this study.  It is among the top most prevalent.

There is surprisingly little cross reactivity among the group 10 mite allergens.  Usually these are highly cross-reactive among all arthropods.  How do the authors explain this?  It may be useful to add a supplementary table showing the sequence identity between house dust mite allergens and Tp allergens.

Table 2 may be better presented as a figure.  The author should consider this.

There are two tables labeled Table 3.  Is this one contiguous table?  The pagination is not clear.

Table 3 (first).  How confident are the authors that Tyr p 8 and Tyr p 20 are not significant allergens?  The prevalence is very close to Tyr p 3.

Table 3 (second) The author show 7 permutation of allergens looking for the highest diagnostic accuracy. Why were these combinations chosen?  Would it be better to test all possible permutations and present a ranking of the few best diagnostic combinations?

The references to mite allergies are all quite old in the discussion. Are there not recent studies on Tp and house dust mites?

Line 296.  Der p I and Der p II is no longer proper allergen nomenclature.

Author Response

Review-3

The paper, “Identification the cross-reactive or species-specific allergens of Tyrophagus putrescentiae and development molecular diagnostic kits for allergic diseases”, describes studies of new and previously identified allergens from T. putrescentiae (Tp) and their applicability to a diagnostic kit.  The paper contains valuable allergological information about mite allergens and cross reactivity with house dust mites.  The Tp species is not as well studied as house dust mites and the paper represents good new information. There are a few major and minor considerations that should be addressed prior to publication.

Major Scientific

Q: A quick look at www.allergen.org reveals that several of these allergens have not been officially named by the WHO/IUIS allergen nomenclature committee.  This must be done prior to acceptance for publication.  Instructions for getting an officially sanctioned name for Tyr p 1, Tyr p 7, and Tyr p 20 can be found at the website and usually takes less than 2 weeks.  This helps keep names correct in the scientific literature and would be a significant contribution to science. 

Reply: We appreciated your valuable and useful suggestions. Such modifications and additional explanations will greatly increase the value of our articles. We fully agree with your comments that the Instructions for getting an officially sanctioned name for these allergens would be a significant contribution to science. However, the experimental data of these allergens are still in progress cooperated with other companies and Academic Subsidy Units. We have not yet obtained the consent of the authors and subsidized units. Especially, one of the authors is a medical student who is studying for a Ph.D. Please forgive our consideration. We will register these allergens and publish the articles as soon as possible, just like the allergens of Tyr p 3 and Tyr p 8.   

 Major Stylistic

Q: The discussion is far too long and could potentially be cut in half.  For example, the paragraphs that begin on line 280 (blood sampling), and 321 (COPD) and not totally applicable or could be cut significantly. The next to last paragraph on the limitations of the study and the conclusion paragraph are the most salient to remain.

Reply: We thank you very much for your suggestion. The discussion on line 280 (blood sampling), and 321 (COPD) in the original manuscript have been deleted. The next to last paragraph on the limitations of the study and the conclusion paragraph are retained.

Q: The authors should utilize a English language professional to correct numerous typos and somewhat silly errors like line 358, “The group 10 mite allergens are involved in music contraction in invertebrates… ” or the nonsensical term in line 99 “IgE fullness”.

Reply: We acknowledge your thoughtful insights and comments. Your suggestions make this article better. The entire article and reply comments have been requested by professional grammar experts to make amendments. Thank you for pointing out this typo. The sentence has been corrected as “muscle” contraction (line 384).

The group 10 mite allergens are involved in muscle contraction in invertebrates and processed high levels of cross-reactivity between mites, shrimp, and insects in seafood-allergic patients, referred to as pan-allergens (50).

The sentence has been corrected as “filled in” (line 103).

It is a possible phenomenon that multiple exposures and co-sensitization to these mite allergens in nature causing free IgE filled in the sera of these allergic patients.

Minor points

Q: Combination D and G+ are named but not explained in the abstract so do not provide valuable information. Please revise to explain succinctly.

Reply: We appreciate for pointing out the insufficient description.  The more detail descriptions about the allergen combination D and H+ (original G+) have been added in the abstract.

Allergenic components had been characterized as cross-reacting or species-specific allergens, in which the effective combinations of recombinant allergens were developed and analyzed the prediction accuracy for clinical diagnosis.

Prediction accuracy for Tp allergy by IgE responsiveness combination D (Tyr p1, Tyr p2 & Tyr p3) was with high precision (100%). Avoiding the cross-reactivity of Dermatophagoides pteronyssinus, the prediction accuracy of IgE responsiveness combination H+ (Tyr p1, Tyr p2, Tyr p3, Tyr p7, Tyr p8, Tyr p10 & Tyr p20) was suitable for Tp-specific diagnosis. Panels of Tp allergens were generated and developed a diagnostic kit able beneficial to identify IgE-mediated Tp hypersensitivity. (lines 33-37)

Q: One is left wondering about the study population in total.  Please provide more demographic information, preferably in tabular format, about the age ranges, severity of diseases, diagnoses etc that might be relevant.  For example, it is confusing in Figure 2, were there no patients 41-64?

Reply: Thank you for reviewing my article very carefully. Indeed, it should be normally distributed of the allergic patient in the clinic. When planning this research in the beginning, we wanted to compare the prevalence differences in allergic patients between these two different age groups. Only the age with less than 40 yrs of young adult subjects and over 65 yrs of elderly subjects were recruited in this study. The more detail information focusing on the frequency of IgE-binding in T. putrescentiae allergenic components in the different age group before and after D. pteronyssinus absorption have been added in the revised Fig 5 and Result section (lines 206-232).

Q: Line 160.  The definition of major allergen is usually prevalence of >50%.  So technically Tyr p 3 is not a major allergen in this study.  It is among the top most prevalent.

Reply: We appreciate for pointing out this incorrect statement. The reviewer is indeed an expert in this field. Thanks for give us sincere and helpful suggestions. The inappropriate description has been corrected as followings in the Result section (Lines 166-169).The major IgE-binding components of T. putrescentiae were group 1 (Tyr p 1) and group 2 (Tyr p 2) with a molecular weight of 25kDa (61.3%) and 16kDa (79.2%). The group 3 allergen (Tyr p 3) with a molecular weight of 26kDa (49.1%) was also one of the most prevalent IgE-binding components.

A: There is surprisingly little cross reactivity among the group 10 mite allergens.  Usually these are highly cross-reactive among all arthropods.  How do the authors explain this? 

Reply: Thank you for your comments. The experimental results of this research are consistent with your cognition and insight. The more detailed descriptions have been added in the sections of Result and Discussion.

In the Result section: (lines 177-181)

The decreased level of IgE-binding was highest in Tyr p 10 (three-fold decreased), following by Tyr p 2, Tyr p 8, Tyr p 20 (two-fold decreased). The level of cross-reactivity between T. putrescentiae and D. pteronyssinus was trivial in Tyr p 1, Tyr p 3, and Tyr p 7. Results showed the highest level of cross-reactivity between T. putrescentiae and D. pteronyssinus were group 10 allergens.

In the Discussion section: (lines 383-386)

In this study, the highest level of cross-reactivity allergen between T. putrescentiae and D. pteronyssinus were the group 10 mite allergens. The group 10 mite allergens are involved in muscle contraction in invertebrates and processed high levels of cross-reactivity between mites, shrimp, and insects in seafood-allergic patients, referred to as pan-allergens [50].

A: It may be useful to add a supplementary table showing the sequence identity between house dust mite allergens and Tp allergens.

Reply: We appreciate your valuable and useful suggestions. The supplementary table has been added in the revised manuscript. The more detail description about the sequence identity and similarity between T. putrescentiae and D. pteronyssinus has been added in the Result section (lines 181-187).

The protein sequences of these allergens were compared between T. putrescentiae and D. pteronyssinus by Basic Local Alignment Search Tool (BLAST), it showed the sequence similarities were over 80% in group 8, group 10 and group 20 allergens (Supplementary Table 1). The lower similarities of protein sequences were respectively in group 1, group 3 and group 7. The analysis of sequence similarity confirmed that it was indeed consistent with the experimental demonstration of cross-reactivity performed by IgE immunoblot inhibition.

A: Table 2 may be better presented as a figure.  The author should consider this.

Reply: We appreciate your valuable suggestion. The original Table 2 has been replaced as Figure 5. This presentation is indeed clearer (Lines 209).  

A: There are two tables labeled Table 3.  Is this one contiguous table?  The pagination is not clear.

Reply: We appreciate for pointing out this error. The mistake had been corrected as Table 2 and Table 3 (line 254 & line 272).

A: Table 3 (first).  How confident are the authors that Tyr p 8 and Tyr p 20 are not significant allergens?  The prevalence is very close to Tyr p 3.

Reply: Thank you for your reminding. The more appropriate description has been revised in the Result section as follows. (lines 169- 171)

The IgE-binding components of group 8 (Tyr p 8) with a molecular weight of 26 kDa (44.3%) and group 20 (Tyr p 20) with a molecular weight of 40 kDa (48.1%) also belonged to significant allergens.

A: Table 3 (second) The author show 7 permutation of allergens looking for the highest diagnostic accuracy. Why were these combinations chosen?  Would it be better to test all possible permutations and present a ranking of the few best diagnostic combinations?

Reply: Yes, we analyzed all possible permutations of allergen IgE responsiveness to find out the most accurate combination set. Because of the size of the table, only a few combinations with high predictive value over 80% of T. putrescentiae allergy were shown in Table 2. The more detail description has been added in the Result and Discussion section.

In the Result section: (lines 246- 248)

All possible permutations of allergen IgE responsiveness were analyzed to find out the most accurate combination set. Because of the size of the table, only a few combinations with high predictive value over 80% of T. putrescentiae allergy were shown in Table 2.

In order to consider the influence of cross-reactivity from house dust mite, the IgE adsorption experiment was performed. The data of IgE responsiveness combinations to species-specific allergens with high predictive value over 55% of T. putrescentiae allergy were shown in the Table 3. (lines 259- 262)

In the Discussion section: (lines 343- 348)

The prediction accuracy for T. putrescentiae allergy by IgE responsiveness combination D (Tyr p 1, Tyr p 2 & Tyr p 3) and E (Tyr p 2, Tyr p 3 & Tyr p 7) could be achieved to 100%. It seems the prediction combinations of these recombinant allergens from major allergens are suitable for diagnostic agents. Recombinant allergens have provided us with high accuracy and efficiency that can improve the diagnosis of allergy.

A: The references to mite allergies are all quite old in the discussion. Are there not recent studies on Tp and house dust mites?

Reply: The updated references have been added as follows in the Discussion section and reference list. (lines 276- 281)

The ubiquitous presence of T. putrescentiae in urban and rural settings is an important occupational hazard in agricultural environments or a common allergy provocation factor in household surroundings (27). In view of existence abundance and high allergenicity, the house dust mite- D. pteronyssinus, D. farinae, and storage mite- T. putrescentiae are at high levels of infestation and associated with airway hypersensitivity for the development of allergic diseases (28, 29).

Most of the allergic patients with mite allergy are almost allergic to both species of mites, D. pteronyssinus and D. farinae, which are the predominant species and coexist in most geographical regions (30). The phenomenon can be speculated that it may be co-sensitized to D. pteronyssinus and D. farinae in the household surroundings or possessed high levels of immunologic cross-reactivity between their allergenic components (30).

Q: Line 296.  Der p I and Der p II is no longer proper allergen nomenclature.

Reply: We thank you for your reminding. The inappropriate nomenclature of allergens has been corrected as follows in the Discussion section (Lines 324).

That is the explanation why the divergences of allergen contents had been mentioned that the levels of both Der p 1 and Der p 2 in D. pteronyssinus extract from the different company reveal variability and the proportion variation of Der p 1 and Der p 2 among different extracts are likely to influence their biological effectiveness [32].

Round 2

Reviewer 3 Report

The authors have addressed many of the initial concerns, and clarified many important points.  However there are two major concerns that must be addressed prior to publication.

  1.  There is something confused regarding the math in section 2.4 and figure 4 that appears incorrect and this affects the conclusions.  The fold change before and after adsorption is stated in the text as 3 fold for Tyr p 10, and 2 fold for Tyr p 8 and Tyr p 20.  However looking at the numbers in the figures seems to suggest there is a 3 fold change for all three allergens.  Which is correct? This must be corrected for consistency.  After correcting for consistency, please also address which are the most important cross reactive allergens. 
  2.  This reviewer will not accept this article for publication unless Tyr p 1, Tyr p 7, and Tyr p 20 are officially named by the WHO/IUIS allergen nomenclature subcommittee at www.allergen.org.  The response time is usually less than two weeks.  The requirements for obtaining a name are significantly less than that required for publication so it should not be a problem with the data already presented or known for this paper.  An acknowledgement from the committee will be required for accepting this paper.

Round 3

Reviewer 3 Report

Thank you for responding to issues in the data and nomenclature.